# Classification of Ancient Roman Coins by Denomination Using Colour, a Forgotten Feature in Automatic Ancient Coin Analysis

**Yuanyuan Ma and Ognjen Arandjelović \*** 

School of Computer Science, University of St Andrews, St Andrews KY16 9AJ, UK
**\*** Correspondence: ognjen.arandjelovic@gmail.com; Tel.: +44-(0)1334-48-26-24

**Abstract:** Ancient numismatics, that is, the study of ancient currencies (predominantly coins), is an interesting domain for the application of computer vision and machine learning, and has been receiving an increasing amount of attention in recent years. Notwithstanding the number of articles published on the topic, the variety of different methodological approaches described, and the mounting realisation that the relevant problems in the field are most challenging indeed, all research to date has entirely ignored one specific, readily accessible modality: colour. Invariably, colour is discarded and images of coins treated as being greyscale. The present article is the first one to question this decision (and indeed, it is a decision). We discuss the reasons behind the said choice, present a case why it ought to be reexamined, and in turn investigate the issue for the first time in the published literature. Specifically, we propose two new colour-based representations specifically designed with the aim of being applied to ancient coin analysis, and argue why it is sensible to employ them in the first stages of the classification process as a means of drastically reducing the initially enormous number of classes involved in type matching ancient coins (tens of thousands, just for Ancient Roman Imperial coins). Furthermore, we introduce a new data set collected with the specific aim of denomination-based categorisation of ancient coins, where we hypothesised colour could be of potential use, and evaluate the proposed representations. Lastly, we report surprisingly successful performances which goes further than confirming our hypothesis—rather, they convincingly demonstrate a much higher relevant information content carried by colour than even we expected. Thus we trust that our findings will be noted by others in the field and that more attention and further research will be devoted to the use of colour in automatic ancient coin analysis.

**Keywords:** colour words; hue histogram; colour representation; machine learning; computer vision

## 1. Introduction

The application of machine learning and computer vision in ancient coin analysis (mostly Ancient Roman coins) is a relatively new but quickly growing area of research [1–3]. Most research to date has focused on what is probably the most obvious and ultimately the important practical task within this field, which can be succinctly summarized by the question asked by a numismatist when presented with a new coin: "What is this coin I've got?". To be clear, the question asked is about the *coin type* [4,5], defined by its semantic content, rather than the identity of the physical specimen itself.

Considering the performance of automatic algorithms in addressing the aforementioned problem in the context of more modern coins, it is probably safe to say that the findings of early research on ancient coins were surprising. In short, the challenge was found to be much more difficult than anticipated [1,6] and the associated performance highly disappointing [7]. While recent work has made

substantial progress, both in terms of technical methodology and the ultimate performance, in this paper, we identify and focus on one particular aspect which all work to date has effectively ignored: colour. In particular, all existing work discards colour and treats coin images as being greyscale. This is not entirely unreasonable given that a coin's type is entirely dependent only on the semantic content depicted on it: the text and the writing on the coin (the so-called legend, in numismatic jargon [8]), the issuing authority shown on the obverse (colloquially, the heads), and the motif on the reverse (colloquially, the tails).

Our counter-argument and indeed the motivation for the present work is manifold. Firstly, considering the challenge that determining a coin's type is proving to be, we observe that a human expert approaches this task in several stages. One of these is the narrowing of possible options by determining the *denomination* of the coin (it is worth remembering that the number of types of Ancient Roman Imperial coins exceeds 43,000 as estimated by Online, http://numismatics.org/ocre/, a joint project of the American Numismatic Society and the Institute for the Study of the Ancient World at New York University). Secondly, we note that the most common denominations used during the existence of the empire, namely dupondii, ases, sestertii, and denarii were made of different materials. As such, the hypothesis we made was that this first, coarse categorization of coins can be done using colour as one of the main features. An additional benefit of this approach lies in its non-reliance on the preservation condition of specimens—the material remains unchanged even if major visual features are destroyed, see Figure 1.

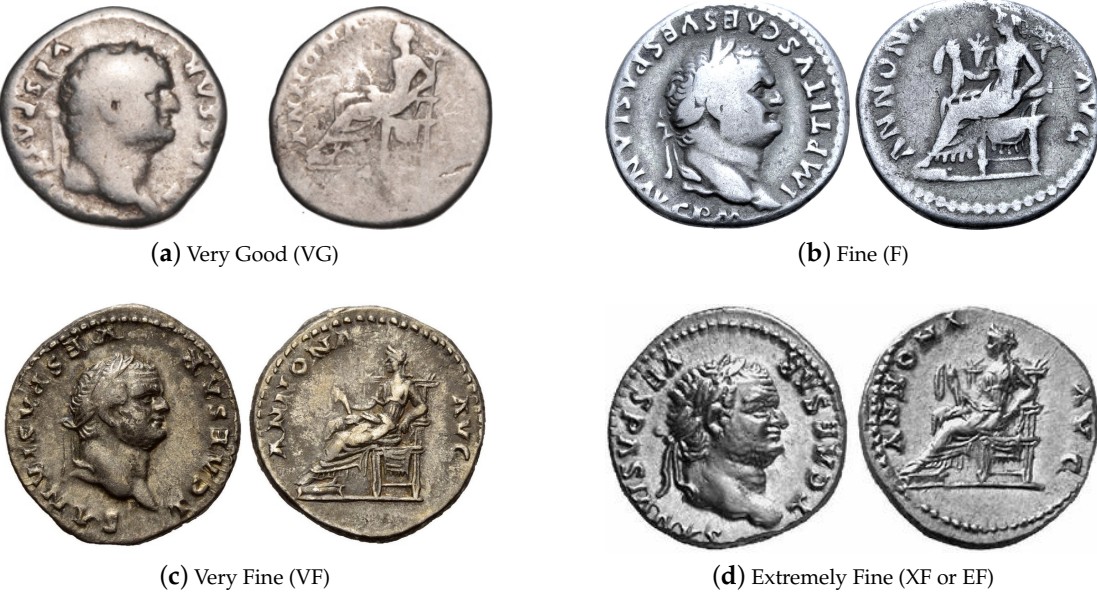

(**a**) Very Good (VG)  (**b**) Fine (F)

(**c**) Very Fine (VF)  (**d**) Extremely Fine (XF or EF)

**Figure 1.** Examples of the same coin type (denarius of emperor Titus; RIC 972 [Vespasian], RSC 17, BMC 319) in different grades of conservation: (**a**) very good (VG), (**b**) fine (F), (**c**) very fine (VF), and (**d**) extremely fine (XF or EF). The two lower grades, namely fair (Fr) and good (G), are not shown due to the lack of interest in specimens damaged so severely.

## 2. How to Represent Colour for Ancient Coin Analysis?

The concept of 'colour' is a rather abstract one and how colour is represented has a major effect on the performance of any learning based on it [9]. The design and choice of representation is also highly domain specific. Herein, for example, we wish to distinguish between ases made of reddish pure copper, dupondii made of golden coloured copper alloy orichalcum, sestertii made of brass, and denarii made of silver, see Figure 2. Notwithstanding the aforementioned differences in the materials, even from this illustration it is clear that visual differences in colour are subtle, especially when it

is taken into account that the exact composition of each denomination varied across time and that various environmental factors effected chemical changes, particularly on coins' surfaces, which can alter appearance significantly.

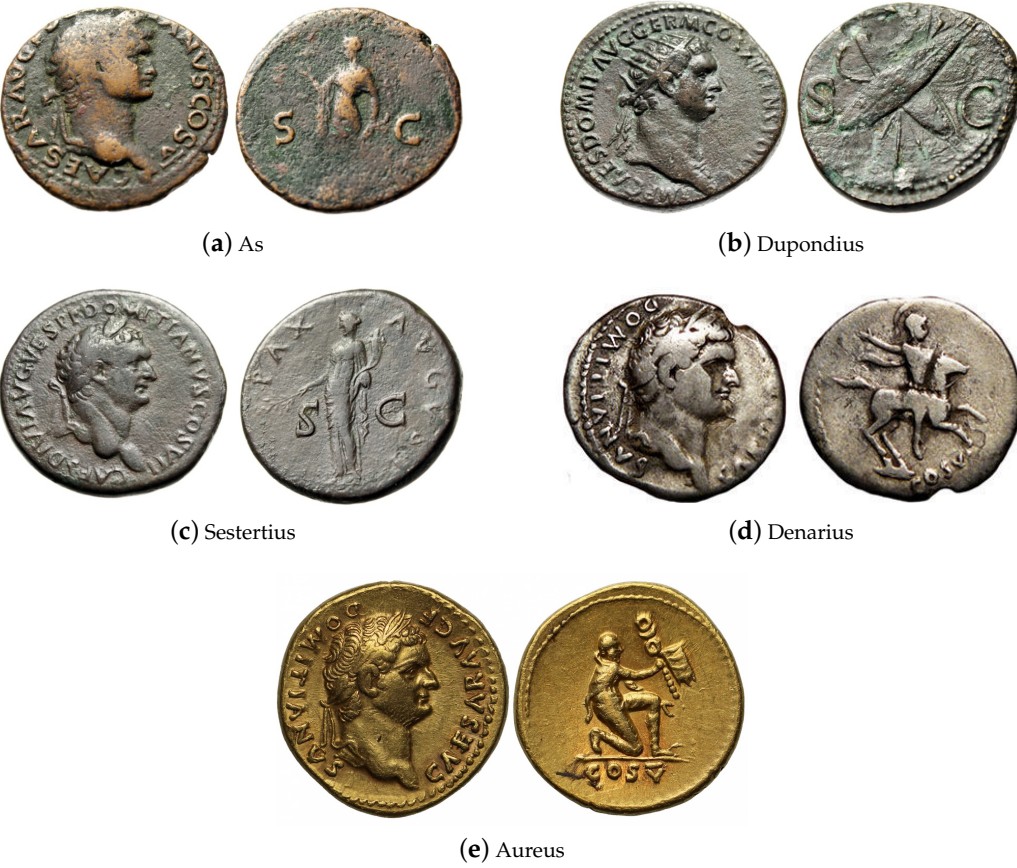

(**a**) As
(**b**) Dupondius

(**c**) Sestertius
(**d**) Denarius

(**e**) Aureus

**Figure 2.** Coins of different denominations of the same emperor, Domitian. Shown are, in order of value at the time of their use, examples of an (**a**) as, (**b**) dupondius, (**c**) sestertius, (**d**) denarius, and (**e**) aureus.

In this section, we propose two different representations, which are then evaluated and compared in the next section.

*2.1. Representation 1: Histogram of Hue*

Our first representation adopts a well-known and widely used colour representation in the form of Hue-Saturation-Value (HSV), which is a simple bijective transformation of the Red-Green-Blue (RGB) space; see Figure 3. In particular, we focus on the hue channel, as the one which contains the inherent information about colour, and discard saturation and value, which are readily affected by extrinsic imaging factors such as illumination strength, duration of exposure, etc. Hue is probably best understood as an angle in the radial HSV space:

$$H = 60° \times \begin{cases} undefined & \text{if } \max(R,G,B) = \min(R,G,B) \\ \dfrac{G-B}{\max(R,G,B)-\min(R,G,B)} \bmod 6 & \text{if } R = \max(R,G,B) \\ \dfrac{B-R}{\max(R,G,B)-\min(R,G,B)} + 2 & \text{if } G = \max(R,G,B) \\ \dfrac{R-G}{\max(R,G,B)-\min(R,G,B)} + 4 & \text{if } B = \max(R,G,B) \end{cases} \quad (1)$$

where *R*, *G*, and *B* are the values of the red, green, and blue components of a pixel respectively, and *H* the corresponding value of hue.

This simple transformation of RGB space allows us to obtain the value of Hue for each pixel in an image of a coin. As a means of reducing noise, that is, the inclusion of unreliable Hue values, we only include in our consideration those pixels which are characterized by significant saturation and value (thresholds of 10 and 200, respectively, given the usual range of 0–255). To create a holistic, robust representation of a coin as a whole, we aggregate all pixels' hue channels into an 8-bit bin histogram. All histograms are thereafter normalized to unit $L_1$-norm which ensures invariance to image resolution and scale.

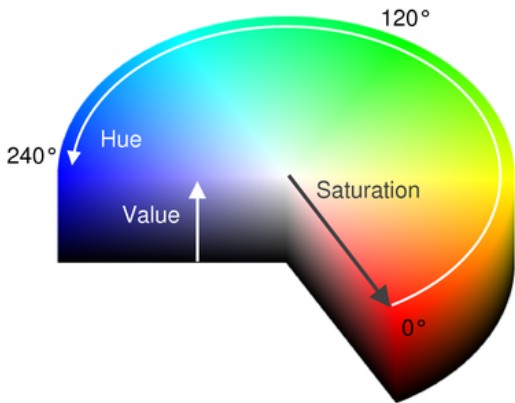

**Figure 3.** Conceptual depiction of the Hue-Saturation-Value (HSV) space at the heart of our first representation.

### 2.2. Representation 2: Histogram of Colour Words

Our second representation is more complex and is motivated by several weaknesses which we identified in the simpler, histogram of hue-based representation introduced first. Conceptually, the main limitation we hypothesised emerges from the lack of data driven specificity—nowhere is the representation *itself* guided by the data it is applied to (that is, ancient coins). Secondly, there is the unappealing aspect of the choice of the bin count—it is difficult to see how a principled choice of the optimal number can be made or indeed why bins would have to be of equal widths (in the hue dimension).

Hence, we propose an alternative, inspired by the successes of the Bag-of-Visual-Words (or related 'bags', such as the Bag-of-Boundaries) in object recognition, categorization, and retrieval tasks [10–12]. To the best of our knowledge, this is the first time this idea has been adopted for use on colour, not only in the domain of computer-based ancient numismatics but in computer vision in general.

The processes of creating and applying histograms of colour words follows the same pattern as those which deal with visual words [13,14]. Firstly, from a training set of images, all pixels, that is, their representations in the 3D RGB space, are collected, and clustering using *k*-means applied. The centres of the resulting clusters become 'colour words' and constitute a colour word dictionary. Thus, an image of a coin is represented as a histogram over this dictionary whereby a count is made of the number of pixels which cluster to each of the dictionary entries. We automatically determine the optimal value of *k* using the minimum description length principle [15].

### 3. Empirical Evaluation

In this section, we evaluate and compare the two proposed representations. We start by describing the new data set collected for this purpose (for research purposes available freely from the authors upon request), then explain our evaluation methodology, and finally present and discuss the obtained results.

### 3.1. Data

For the evaluation of the proposed methods, we collected a new data set of 400 coins (that is, images thereof), with 100 exemplars for each of the four denominations of interest; recall these are ases, dupondii, sestertii, and denarii. To ensure the absence of confounding factors not of interest herein (clutter, variable background, etc. [16]), allowing us to gain insights specifically into the hypothesised usefulness of colour and our representations in particular, all coins are imaged against a uniform background. Representative examples are shown in Figures 4 and 5.

### 3.2. Experimental Methodology and Details

Considering the focus of the present work, that is, the examination of the relevant information content of the modality (namely colour) under consideration and our different approaches to the representation thereof, as well as the lack of any prior investigation of the same in the existing literature, we decided to opt for a simple and readily interpretable classification approach in the form of random forests [17] rather than one of the recent deep-learning-based methods, despite their recent successes in this domain [18–22]. We used a forest consisting of 30 random trees.

As noted previously, we used the minimum description length principle [15] to infer automatically the optimal value of clusters (colour words) for our colour-word-based representation, which resulted in a compact dictionary with $k = 6$.

Lastly, to ensure a lack of bias in our results, we adopt the usual $n$-fold validation procedure with $n = 10$. Moreover, the validation is repeated 2000 times for additional robustness of findings.

### 3.3. Results

Recall that our hypothesis was that colour is a useful feature which could prove valuable in coarse, first stage categorization of Ancient Roman Imperial coins, not the *sole* feature. Our expectation and hope were that colour-based features would prove informative but certainly not very discriminative. Hence we were rather astonished to discover just how good the performance of even our first, hue-histogram-based representation introduced in Section 2.1 was. It is illustrated well by the confusion matrix shown in Table 1. Firstly, note that the performance of denarii and ases matching is outstanding, comfortably exceeding 90% for both types. Best performance for denarii was expected as their composition (silver) sets them quite apart from the other denominations, but the success rate of this magnitude was surprising. Even more surprising was the accuracy of matching ases given their copper-based composition, which is not that much unlike the composition of dupondii and sestertii. The matching performance for these latter two denominations was worse but still higher than we expected. What is more, they were for the most part confused one with the other which is reassuring given their similar compositions, demonstrating that our features are capturing the desired colour characteristics well.

**Table 1.** Confusion matrix corresponding to our histogram of hue-based representation and a random forest classifier (entries in bold denote correct classifications). N.b. The numbers in some rows do not add up to 1 due to rounding errors.

| True/Predicted Class | As | Denarius | Dupondius | Sestertius |
|---|---|---|---|---|
| **As** | **0.92** | 0.01 | 0.02 | 0.06 |
| **Denarius** | 0.00 | **0.91** | 0.04 | 0.05 |
| **Dupondius** | 0.03 | 0.05 | **0.59** | 0.33 |
| **Sestertius** | 0.05 | 0.11 | 0.29 | **0.55** |

Further insight into the performance of our representation can be gained by examining the corresponding Cumulative Match Characteristic (CMC) curve, shown in Figure 6. The average accuracy across all four denominations can be seen to shoot up already at rank-2 which corroborates our previous observations.

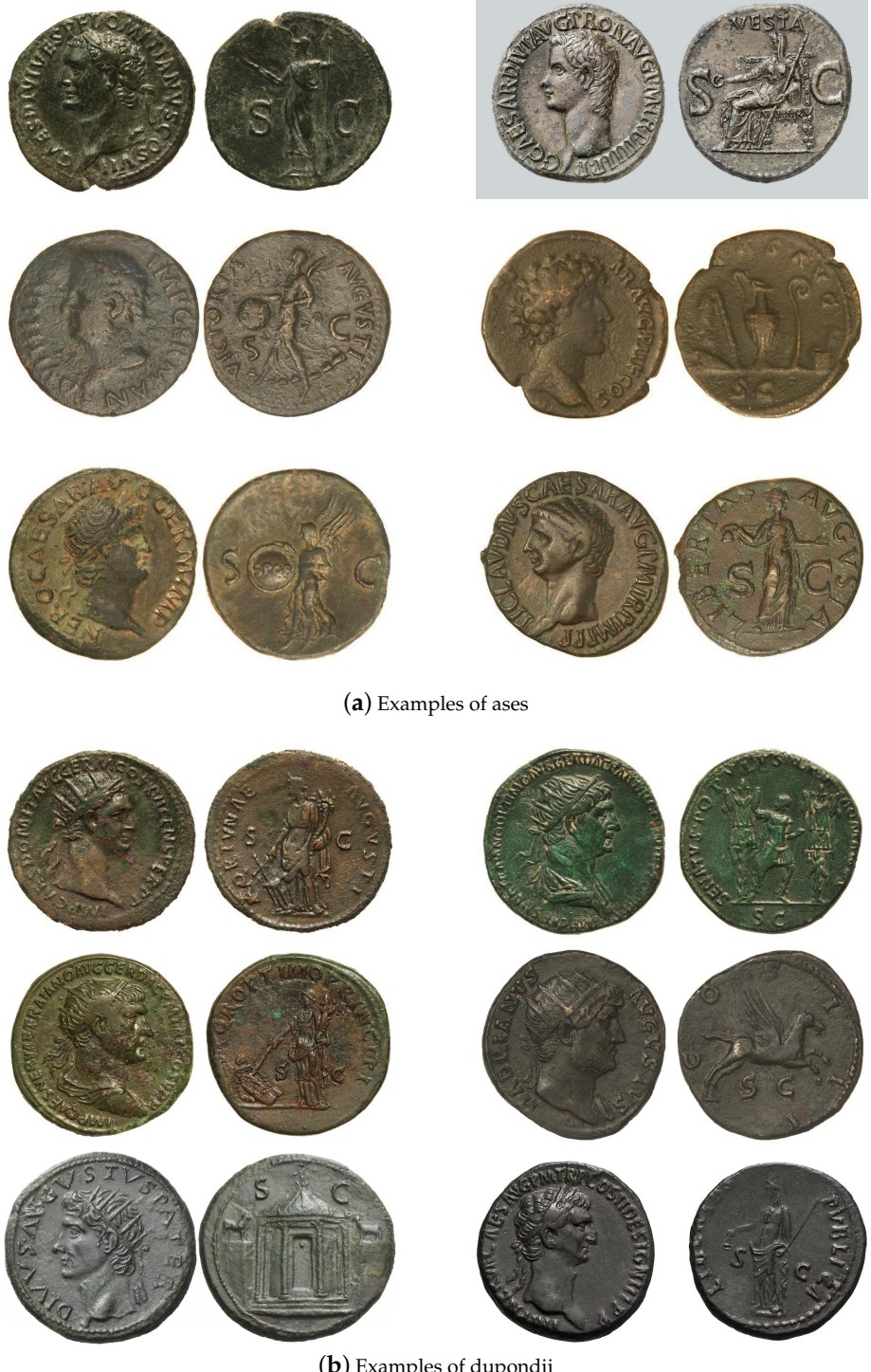

(**a**) Examples of ases

(**b**) Examples of dupondii

**Figure 4.** Examples of (**a**) ases and (**b**) dupondii in our data set. Also see Figure 5.

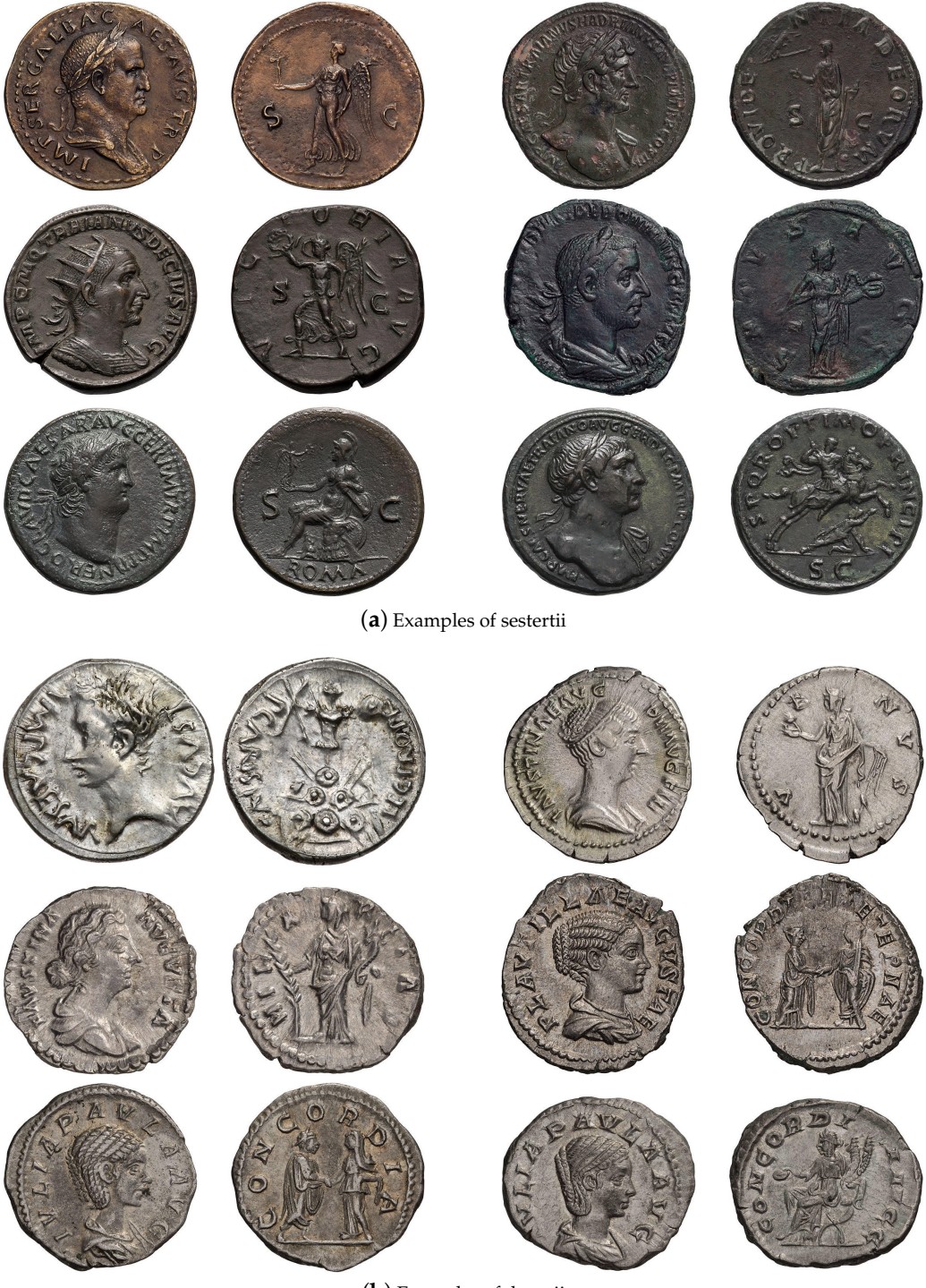

(**a**) Examples of sestertii

(**b**) Examples of denarii

**Figure 5.** Examples of (**a**) sestertii and (**b**) denarii in our data set. Also see Figure 4.

It is interesting next to compare our first, hue-histogram-based representation with the more complex, colour-word-based one described in Section 2.2, and observe the similarities and differences in behaviour. The results are summarized in Table 2. The foremost similarity, which too was surprising to us at first (although not in the light of the results presented thus far) was again its better than expected performance for such a simple feature. The accuracy of categorizing ases and denarii was again high at nearly 90% (thus somewhat worse than with the previous method), but now, the differentiation between dupondii and sestertii was much improved over the hue-histogram-based approach—the corresponding error rate was nearly halved.

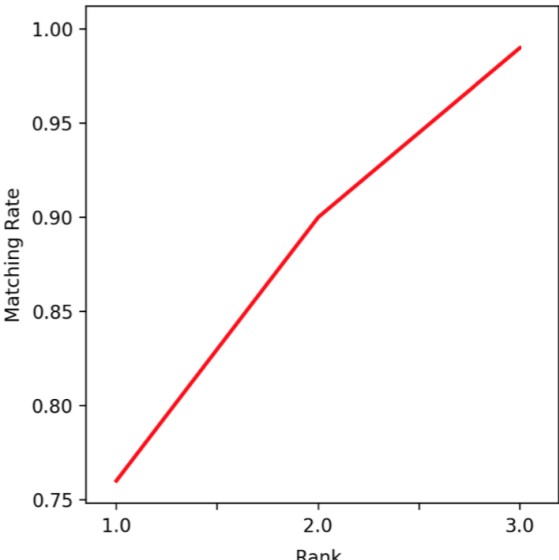

**Figure 6.** The cumulative match characteristic (CMC) curve obtained with our hue-histogram-based representation and a 30 tree random forest classifier.

**Table 2.** Confusion matrix corresponding to our histogram of colour-word-based representation and a random forest classifier (entries in bold denote correct classifications). N.b. The numbers in some rows do not add up to 1 due to rounding errors.

| True/Predicted Class | As | Denarius | Dupondius | Sestertius |
|---|---|---|---|---|
| **As** | **0.89** | 0.04 | 0.02 | 0.05 |
| **Denarius** | 0.00 | **0.89** | 0.07 | 0.03 |
| **Dupondius** | 0.02 | 0.09 | **0.64** | 0.25 |
| **Sestertius** | 0.04 | 0.06 | 0.16 | **0.73** |

The observed differences in the ultimate categorization outcomes on the one hand, and the highly successful performance of both representations on the other, motivated our last experiments which involved the fusion of the two approaches. Thus, we apply simple score-level combination [23] of two random forests (one trained for hue-histogram-based representation and one for colour-word-based one) in the form of quadratic mean-based fusion [24]. As before, our findings are summarized in the form of the corresponding confusion matrix in Table 3.

**Table 3.** Confusion matrix corresponding to our score-fusion-based method which combines our histogram of hue-based and histogram of colour-word-based representations, both used with a random forest classifier (entries in bold denote correct classifications). N.b. The numbers in some rows do not add up to 1 due to rounding errors.

| True/Predicted Class | As | Denarius | Dupondius | Sestertius |
|---|---|---|---|---|
| **As** | **0.96** | 0.01 | 0.01 | 0.01 |
| **Denarius** | 0.00 | **0.96** | 0.02 | 0.02 |
| **Dupondius** | 0.01 | 0.02 | **0.81** | 0.15 |
| **Sestertius** | 0.02 | 0.00 | 0.14 | **0.84** |

## 4. Summary and Future Work

In this article, we made a number of important contributions to the field of computer vision and machine-learning-based analysis of ancient coins. The contributions are varied in nature and include conceptual, technical, and practical novelties of significance. Firstly, despite the large number of publications in this domain, this is the first work to recognise the potential loss of salient information associated with the lack of consideration of colour in analysing ancient coins. We discuss this at some length, recognising the reasons for this choice but argue against it by presenting an argument which uses both inspiration from the manner in which human expert numismatists approach the task and objective inspection of the problem. This in turn motivates our technical contributions in the form of two novel colour-based representations of ancient coins in images. The first of these is based on the distribution of hue across pixels in an image, whereas the second one introduces what we term 'colour words' in analogy to 'visual words' extensively used in computer vision with outstanding success. We also make a practical contribution to the community by collecting a data set curated in a manner which allows for our hypothesis to be tested, and indeed which should facilitate others' further work. Lastly, we present empirical results which demonstrate the validity of our arguments, the power of the proposed representations, and succinctly, performance which vastly exceeded even our hopes for such a simple modality which we expect to be used in a multi-modal framework.

*Future Work*

Our contributions open a number of different avenues for future work. Firstly, as we noted, we intentionally used a simple classifier because our focus was on the modality under consideration (colour) and the proposed representations thereof. Considering that our hypothesis has been confirmed convincingly, it is sensible to explore the use of more complex machine learning approaches, such as deep learning.

It is also worth exploring improvements to our hue-histogram-based representation. In particular, in this histogram, unlike in those based on dictionaries (such as the dictionary of colour words), neighbouring histogram bins are semantically related. Thus, it is sensible to explore means of incorporating this knowledge as a means of reducing noise in empirically estimated distributions (e.g., the simplest approach could simply apply Parzen window estimation [25]).

Lastly, it is without doubt a good idea to explore different ways of performing feature-level rather than score-level fusion, which is both a more principled approach to the task and highly likely a practically better one considering the relatively low dimensionality of the problem at hand.

**Author Contributions:** All authors have contributed to all aspects of the described work. All authors have read and agreed to the published version of the manuscript.

**Funding:** This research received no external funding.

**Conflicts of Interest:** The authors declare no conflict of interest.

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
