# Peer review of "Classification of Ancient Roman Coins by Denomination Using Colour, a Forgotten Feature in Automatic Ancient Coin Analysis"

_sci, doi:10.3390/sci2020037_

Round 1

Reviewer 1 Report

Comments on “Classification of Ancient Roman Coins by Denomination Using Colour, a Forgotten Feature in Automatic Ancient Coin Analysis”

Punctuation errors

Incorrect subject/verb agreement

Erroneous spacing between words

Some phrases (and spelling) are arcane and do not facilitate meaning.

For a reader who is not well-versed in statistics, the article would be confusing.

Authors do not provide enough information for an informed lay person (and possibly a trained numismatist) to understand how the determination of identification of type was reached on the basis of a histogram of color words when an example of the histogram with its range of color words is not provided. 

Authors stress that color is only one parameter by which a coin type might be assigned, but one wonders whether dimensions and weight could be considered as well.  There is a relatively specific range of diameter for specific coin types, and the same could be said for weight.

Are all the coin images presented in the article shot at the same scale?  This is not specified.

It is not at all clear to me (an archaeologist) how the number derived from the formula on page 3 of the article translates into specific words for hue (and its variation) that form the basis of a histogram of color words.

I must say that the premise of the article is an intriguing one and any numismatist (or archaeologist) would be interested in learning more about an approach that permits the variation of specific hue to be accounted for but still allows for a validity of identification of coin types based on color.  I found that there was not enough explanation and description of the methodology that was presented in a way that an informed archaeologist could understand, and this detracted from my own professional assessment of the article. 

The authors did not comment on whether all the coins that were assessed were derived from the same mint (one would think that this would impact on the variation of the metal content) nor did they address whether commonalities of the dates of issue were taken into consideration.

Author Response

Reviewer 1's comments. We are absolutely appalled by the poor intellectual standard of this review.  The reviewer is clearly unsuitable for reviewing articles in this field, and yet appears to be oblivious of the importance of this fact. To start with, consider the following statement: "For a reader who is not well-versed in statistics, the article would be confusing."  Indeed, we are sure that is so and such readers are indeed not our target audience.  We are talking about a technical article in an academic journal and the specific special issue entitled "Machine Learning and Vision for Cultural Heritage".  Is it not clear to the reviewer that the understanding of statistics is crucial for understanding contributions in this area?  How can the reviewer possibly justify her acceptance to review in this field without such basic technical knowledge? The reviewer's staggering display of unsuitability as reviewer in this field continues: "...one wonders whether dimensions and weight could be considered as well.  There is a relatively specific range of diameter for specific coin types, and the same could be said for weight."  Our manuscript deals with *computer vision* based analysis of coins - only images are available.  That this would need explaining really leaves one speechless. Then we have more of the same: "Authors do not provide enough information for an informed lay person (and possibly a trained numismatist) to understand how the determination of identification of type was reached on the basis of a histogram of color words..."  Again, this paper is NOT aimed at "informed lay people" or numismatists but rather researchers in the field of computer vision based analysis of ancient coins.  The clue is in the publication venue: it is not a pop science magazine, nor a numismatics journal, but a scientific journal and the specific special issue entitled "Machine Learning and Vision for Cultural Heritage".  The concepts the reviewer brings up are familiar to just about everybody with a basic level of competence in the field, which again highlights the inappropriateness of this reviewer. Predictably, the same lack of elementary knowledge continues: "It is not at all clear to me (an archaeologist) how the number derived from the formula on page 3 of the article translates into specific words for hue (and its variation) that form the basis of a histogram of color words." These are such elementary things that they the reviewer's feedback really requires no further comment.  We are just amazed at the reviewer's lack of self-reflection in the decision to accept to review our article. The same kind of issues are repeated thereafter so we will refrain from commenting on each individually. Lastly, as to "arcane" wordings, we are indeed aware that properly spoken or written English can sound "arcane" to many, e.g. to those who think that the sentence "Some phrases (and spelling) are arcane and do not facilitate meaning." makes any sense (it is e.g. "better understanding", not "meaning", that the choice of phrases can facilitate). Any remaining typos (and we are sure that some likely do remain, though they should have been corrected by the publishing office staff) we would have been glad to have highlighted to us, so that we can correct them.  Unfortunately, the reviewer's lack of professionalism is shown here too as no specifics are provided.  Spacing and similar layout issues are not under our control - the manuscript's final layout is done by the publisher as the reviewer ought to know.

Reviewer 2 Report

Koerner review of sci-746995
Title: Classification of Ancient Roman Coins by Denomination Using Colour, a
Forgotten Feature in Automatic Ancient Coin Analysis
by  Yuanyuan Ma, Ognjen Arandjelovic *
for special issue if SCI - Machine Learning and Vision for Cultural Heritage

This article is a strong mix between potential and difficulties in terms of clarity of aims - approaches - findings and conclusions.

potential

The themes and purposes of this article are extremely interesting and have considerable bearing upon the novel aims of the Special Issue

e.g., developments in computer "machine" based "learning" (knowledge acquisition, production, distribution, use) and research in cultural heritage

in these connections - it bears noting extraordinary possible comparisons between themes of the article - and the Special Issue and key themes at the heart of connections in early modern times between innovations in the production, use and distribution of printed texts - and, especially, engraved images and innovations in the roles of the extremely ancient field of numismatics in the dynamics of antiquarian and historical study of the past - with attention to "cultural heritages"

it also bears noting the potential of the extent to which the emphasis the article places on "colour" as a significant line numismatic evidence parallels comparable arguments at the heart of the roles the latter innovations played in deep and far reaching - even 'revolutionary' change in historical reasoning and humanities centring on material culture.

difficulties

There are, however, difficulties in terms of clarity of aims - approaches - findings and conclusions.  These are further complicated by the extent to which key sections lack general orienting details needed to bring relevant points into relief.

suggestions

Addressing this problem - along with general writing problems (complex sentences, grammar etc) might help bring valuable insights and findings into relief -- and with attention to relevance for the aims of the Special Issue.

Author Response

> The themes and purposes of this article are extremely interesting and have considerable bearing > upon the novel aims of the Special Issue e.g., developments in computer "machine" based "learning" > (knowledge acquisition, production, distribution, use) and research in cultural heritage We are very grateful to the reviewer for their time, kind words, and constructive feedback. It goes without saying that we took suggestions on the clarity of certain portions of the manuscript constructively and have thus revised the presentation in line with these. Thank you!

Reviewer 3 Report

The suggested methods are definitely interesting both for computer scientists and for archaeologist, but at the present state they can only be understood by scholars familiar with these particular issues. They could have been described in a more extensive and explicative way in order to be as accessible as possible to specialists in numismatics, and maybe also to the general public. For this purpose, terminology needs to be elucidated and processes need to be detailed and formulated in simple terms. It should also be made clear for non-specialists why those tools are preferable to previous proposals and allow to obtain better results.

Moreover, the proposed application of these color-based representations does not correspond to the best way to take advantage of them. Their use for classification purposes raises many questions, as colors are a changeable factor which cannot be considered as a permanent and strictly defined characteristic of a specific category of coins. The described procedures could be rather used for the depiction of coins in scientific papers, but also in exhibitions, catalogues, books for a wider readership etc. They can also be included as additional indicators in databases, in order to study and compare specific objects in relation to their history and archaeological context. The aims of the paper should therefore be reviewed so as to provide a more relevant connection with the needs and particularties of numismatics. The related parts of the text should be thoroughly rewritten.

Author Response

> The suggested methods are definitely interesting both for computer scientists and for archaeologist We are genuinely thankful for the reviewer's time and thorough constructive feedback. > at the present state they can only be understood by scholars familiar with these particular issues We do agree with this but please note that this is a special issue precisely aimed at this audience. References to relevant published work which the reader can consult if they need to be familiarized with the field are provided - such content is not exactly within the scope of our contribution. We trust that the reviewer will agree. > colors are a changeable factor which cannot be considered as a permanent and strictly defined > characteristic of a specific category of coins Indeed, and this is exactly why machine learning is employed. It is by virtue of learning from examples that the scope and nature of the variations that the reviewer mentions are learnt - they are not arbitrary, unpredictable variations but constrained by the material, environmental factors, etc. Our empirical results demonstrate this.

Reviewer 4 Report

Dear Aimie,

thanks. This is certainly a difficult issue, as there is a disagreement between the authors and the reviewers.

In the case of a traditional journal, I would in such a case recommend the involvement of an additional reviewer whose task it is to focus on the disagreement and to resolve the situation. This often helps and if not, the manuscript is then rejected. An additional reviewer is always an option.

In Sci, this is rather difficult as we have now several reviews already and also little space and scope of another review(er).

I guess in this case, we simply fix the manuscript now as published version and close the process then. It is unlikely that the authors are changing the manucript further, and they - and the readership - then need to accept that this is the version produced during the post-publication peer review.

I personally have no problem with this, as the comments of the reviewers and also the response of the authors are clearly visible and anyone reading this manuscript can then decide if and how valuable this publication, in its different versions, really is.

In any case, thanks again for your patience and let's move on with this manuscript.

Best wishes,

Claus

Round 2

Reviewer 2 Report

looks better - ok - bring key points more into relief